# Characteristics of patients with longer treatment period of lenvatinib for unresectable hepatocellular carcinoma: A post-hoc analysis of post-marketing surveillance study in Japan

**Tatsuya Yamashita**[1] *, **Natsumi Suzuki**[2], **Katsuaki Motoyoshi**[2], **Wanjun Zhu**[2], **Junji Furuse**[3]

1 Department of Gastroenterology, Kanazawa University Hospital, Kanazawa, Japan, 2 Eisai Co., Ltd., Tokyo, Japan, 3 Department of Gastroenterology, Kanagawa Cancer Center, Yokohama, Japan

* ytatsuya28@gmail.com

**Data Availability Statement:** Individual data were provided to Eisai under a contract between the participating institutions and Eisai. Due to

## Abstract

Patient profiles suitable for long-term lenvatinib treatment for unresectable hepatocellular carcinoma (uHCC) are yet to be fully understood. This post-hoc analysis aimed to identify such patient characteristics and explore the impact of treatment duration and relative dose intensity (RDI) on treatment outcomes. The data were obtained from 703 patients in a multi-center, prospective cohort study in Japan. Lenvatinib-naïve patients with uHCC were enrolled between July 2018 and January 2019 and were followed up for 12 months. Moreover, patients were dichotomized using the median treatment duration into the longer- ($\geq$177 days; n = 352) or shorter-treatment (<177 days; n = 351) groups. The longer-treatment group often had better performance status, lower Child-Pugh score and better modified albumin-bilirubin grade than the shorter treatment group (p<0.05 for all). The objective response rate (47.6% vs. 28.2%; p<0.001) and disease control rate (92.4% vs. 60.2%; p<0.001) were both significantly higher in the longer-treatment groups than in the shorter-treatment groups. The proportion of patients with any adverse drug reactions was generally similar between the two treatment groups. Within the longer-treatment group, the disease control rate was high regardless of dose modification (i.e., RDI <100% vs. $\geq$100% during the initial 177 days) (91.2% vs. 98.0%). In conclusion, patients with longer treatment tended to have better overall conditions. Lenvatinib dose modifications at the physician's discretion, considering the balance between effectiveness and safety, may contribute to the long-term treatment.

## Introduction

Liver cancer mortality has been gradually declining in Japan [1], owing to the improvement in its treatments. However, hepatocellular carcinoma (HCC) has a high recurrence rate and

restrictions by Eisai, the underlying data may not be made publicly available. However, data are available to interested and qualified researchers who meet the criteria for access to confidential data upon request to the Clinical Trial Disclosure site of Eisai. For further information, please access the following link: https://www.eisai.co.jp/innovation/research/clinical_trials/clinical/index.html (Japanese); https://www.eisai.com/innovation/research/clinical_trials/clinical/index.html?_gl=1*801ekg*_ga*MjI5NDMwMjY3LjE2OTgzNjcwNzE.*_ga_X1FWS6YR87*MTY5ODM2NzA3MS4xLjAuMTY5ODM2NzA3OC41My4wLjA (English).

**Funding:** This study was funded by Eisai Co., Ltd. The sponsor was involved in the study design, data collection and analysis, decision to publish, and preparation of the manuscript.

**Competing interests:** I have read the journal's policy and the authors of this manuscript have the following competing interests: Tatsuya Yamashita has received travel grants and honoraria for speaking or participation at meetings from Chugai Pharma, Eli Lilly Japan, and Eisai. Natsumi Suzuki, Katsuaki Motoyoshi, and Wanjun Zhu are employees of Eisai Co., Ltd. Junji Furuse received research grants from Astellas, Astra Zeneca, Incyte Biosciences Japan, Eisai, MSD, Ono Pharmaceutical, Sanofi, J-Pharma, Daiichi Sankyo, Sumitomo Dainippon, Taiho Pharmaceutical, Takeda, Delta-Fly-Pharma, Chugai Pharma. He has received travel grants and honoraria for speaking or participation at meetings from Ono Pharmaceutical, Chugai Pharma, Incyte Biosciences Japan, Eisai, Eli Lilly Japan, Astra Zeneca, Yakult Honsha, Servier Japan, MSD, Novartis Pharma, Takeda, Bayer, Taiho Pharmaceutical, EA Pharma, Teijin pharma, Daiichi Sankyo, Terumo, Fuji film, Astellas, Onco Therapy Science, Delta-Fly-Pharma, Merck Bio, Incyte Biosciences Japan, J-Pharma. This does not alter our adherence to PLOS ONE policies on sharing data and materials.

frequently progress to advanced stages of HCC, which affects patient survival. Systemic therapy is an important treatment option for such advanced HCC that is unamenable to surgical resection or locoregional therapy [2]. In Japan, guidelines recommend that the first-line treatment for unresectable HCC (uHCC) is atezolizumab plus bevacizumab if indicated; otherwise it is sorafenib or lenvatinib [2].

Lenvatinib is an oral multi-kinase inhibitor that exerts antitumor effects by inhibiting angiogenesis, selectively inhibiting receptor tyrosine kinases such as vascular endothelial growth factor receptors 1–3, fibroblast growth factor receptors 1–4, platelet-derived growth factor receptor α, RET, and KIT [3]. Lenvatinib was approved in Japan in 2018 for the first time in the world, based on a global, phase 3 clinical trial that showed its non-inferiority to sorafenib [4]. Since then, real-world safety and effectiveness data of lenvatinib for uHCC have been accumulated [5, 6], and it is now widely used in clinical practice.

To date, a few small studies have suggested that patients with better baseline liver functional reserve are more likely to receive longer lenvatinib treatment [7, 8]. However, patient profiles suitable for extended treatment have yet to be fully understood. Such information is essential to identify patients who will benefit the most from lenvatinib treatment. Meanwhile, a positive correlation was observed between the duration of treatment with molecular targeted agents and overall survival (OS) (i.e., time to death) [9].

In lenvatinib therapy for uHCC, an objective response has been reported to be an independent predictor of OS [10], and it has been suggested that a high relative dose intensity (RDI) during the initial treatment phase is important for achieving a tumor response [11–15]. However, a high dose can also increase the risk of treatment discontinuation due to adverse events (AEs) [14], thus hindering longer treatment.

The present study aimed to characterize patients who could be treated with lenvatinib for extended period. We described the patient characteristics and analyzed the treatment outcomes, according to the treatment duration (longer or shorter). We also explored the impact of the RDI on the effectiveness and safety of lenvatinib, as little is known about its impact on the treatment outcomes, particularly after the initial phase.

## Methods

### Study design and patients

This post-hoc analysis examined patient characteristics and treatment outcomes according to the treatment duration or RDI, using the data from a prospective, multicenter, observational post-marketing study of lenvatinib in patients with uHCC in Japan (ClinicalTrials.gov Trial Registration ID: NCT03663114). The study was conducted as part of the pharmacovigilance activities as required by the Japanese Pharmaceutical and Medical Devices Agency (PMDA). Therefore, the Japanese authority, PMDA, reviewed and approved the study protocol, and no ethics approval was required for the study. However, in cases where assessment by the ethics committee or Institutional Review Board of a clinical site was deemed necessary, this study protocol was reviewed. Under the Good Post-Marketing Study Practice (GPSP), patient data are collected at medical institutions in a manner that ensures personal information cannot be identified. Accordingly, in principle, informed consent was not required. However, written or oral informed consent was obtained from all participants at each participating institution, and consent was documented at each institution regardless of how the consent was obtained. The study was conducted in accordance with the provisions of the Declaration of Helsinki, the Pharmaceutical Affairs Law and the Ministerial Ordinance on the GPSP in Japan.

Detailed study methods have been published recently [6]. In brief, the study enrolled lenvatinib-naïve patients with or without any prior history of anti-angiogenic agent treatment who

agreed to participate in the post-marketing study between July 2, 2018 and January 25, 2019. The patients were followed-up for 12 months from the first administration of lenvatinib. Lenvatinib (Lenvima®; Eisai Co., Ltd., Tokyo, Japan) was orally administered once daily. The standard dose was 12 mg/day for patients with body weight ≥60 kg and 8 mg/day for those with body weight <60 kg; however, doses could be adjusted at physician's discretion. The present analysis used data from all of the 703 patients, which comprised the safety analysis set of the original study.

### Study outcomes and analysis sets

In this analysis, we evaluated (1) patient characteristics and treatment effectiveness and safety according to treatment duration and (2) treatment effectiveness and safety with or without dose modification in patients with longer-treatment durations. For (1), the 703 patients were dichotomized based on the median treatment days as the cut-off value: longer-treatment (≥177 days) or shorter-treatment (<177 days) groups (Fig 1). For (2), our intention was to explore the impact of dose modification for a relatively long period; patients in the longer-treatment group were stratified by either with or without dose modification (i.e., RDI <100% or RDI ≥100%) during the initial 177 days of treatment.

### Baseline patient variables

The following baseline variables were assessed to describe the demographic and clinical characteristics of the patients: sex, age, body weight, body mass index, Eastern Cooperative Oncology Group performance status (ECOG PS), Barcelona Clinic Liver Cancer (BCLC) stage, portal vein invasion, Child-Pugh score, modified albumin-bilirubin (mALBI) grade [16],

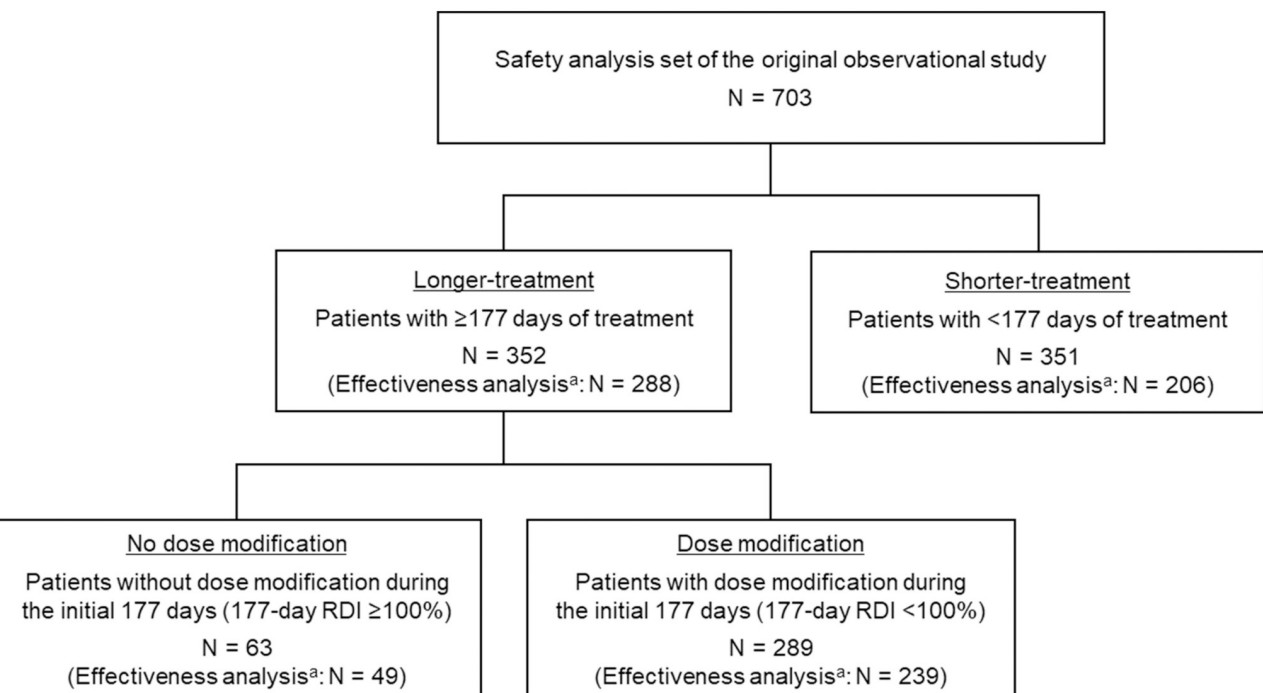

**Fig 1. Flow diagram of analysis populations.** [a]Tumor response was assessed in patients with tumor evaluation data using the mRECIST criteria. RDI, relative dose intensity.

etiology of HCC, history of tyrosine kinase inhibitor (TKI) treatments, history of trans-arterial chemoembolization (TACE), and baseline alpha-fetoprotein (AFP) level.

### Assessment of tumor response

Treatment effectiveness was evaluated in terms of tumor response. Tumor response was assessed by attending physicians using imaging data in three time periods: (1) from treatment initiation to 3 months, (2) from 3 to 6 months, and (3) from 6 to 12 months. In this study, the tumor response was evaluated using the modified Response Evaluation Criteria in Solid Tumors (mRECIST) [17]. Patients who were assessed using any other criteria were excluded from the analysis of treatment effectiveness. According to the best treatment response using the mRECIST criteria, the objective response rate (ORR) was defined as the proportion of patients with either a complete response (CR) or a partial response (PR). The disease control rate (DCR) was defined as the proportion of patients with objective response (CR or PR) or stable disease (SD).

### Assessment of adverse drug reactions

For safety, adverse drug reactions (ADRs) were assessed. In addition to any ADRs, the occurrence of the following most common ADRs identified previously in the overall population (incidence proportions >10%) was examined: appetite loss, fatigue, hypertension, proteinuria, hand-foot skin reaction (HFSR), hypothyroidism, and diarrhea [6]. ADRs were defined as AEs for which a causal relationship with lenvatinib could not be denied. The ADRs were assessed from the start to 14 days after the last administration of lenvatinib. ADRs were classified according to the Japanese version of the Medical Dictionary for Regulatory Activities (MedDRA/J) version 23.1, and their severity was graded according to the Japanese version of the Common Terminology Criteria for Adverse Events (CTCAE) version 4.0.

### Statistical analyses

Patient characteristics, tumor responses, and ADRs were descriptively summarized for each group. The chi-square test was used to compare frequencies between two groups when appropriate, whereas Fisher's exact test was performed to compare tumor responses. For exploratory purposes, we additionally analyzed treatment duration and OS by tumor response (CR/PR vs. SD/PR) using the Kaplan-Meier methods and compared them using the log-rank test. OS was defined as the time from treatment initiation to death. Statistical significance was set at a p-value of <0.05. All p-values are nominal and not adjusted for multiple testing, thus are to be taken with caution. All statistical analyses were performed using the Statistical Analysis System (SAS) Release 9.4 (SAS Institute Inc., Cary, NC, USA).

## Results

### Baseline patient characteristics according to treatment duration

The analysis included all 703 patients from the previously reported observational study. Men were predominant (80.2%), and 83.4% were ≥65 years (40.3% were 65–74 years, 36.3% were 75–84 years, and 6.8% were ≥85 years). The complete patient characteristics and the effectiveness and safety data of these 703 patients have been previously published [6]. The median treatment duration was 177 days (range: 2–482 days).

We dichotomized these 703 patients according to the median treatment duration of 177 days (longer- and shorter-treatment groups, n = 352 and 351, respectively) (Fig 1). The median treatment duration for the longer- and shorter-treatment groups was 366 days (range: 177–482

days) and 70 days (range: 2–174 days), respectively. The baseline characteristics of the patients in each group are summarized in Table 1. The following baseline variables were distributed differently between the two groups with a p-value of <0.05: age, ECOG PS, Child-Pugh score, mALBI grade, history of TKI treatments, portal vein invasion, and AFP level. Compared with the shorter-treatment group, the longer-treatment group consisted of more patients aged <75 years (63.4% vs. 50.4%), and an ECOG PS of 0 (83.8% vs. 67.8%) and fewer patients with portal vein invasion (18.5% vs. 28.5%), no history of TKI treatments (78.4% vs. 84.3%), and AFP levels of ≥200 ng/mL (28.4% vs. 39.9%). The majority of patients in the longer-treatment group had a Child-Pugh score of 5 (65.9%) and mALBI grade 1 or 2a (70.5%). In contrast, the corresponding proportions were 45.9% and 47.6% in the shorter-treatment group. Other variables such as body weight and BCLC stage were similarly distributed between the two groups.

## Effectiveness and safety of lenvatinib according to treatment duration

**Tumor response.** Of the 352 and 351 patients in the longer- and shorter-treatment groups, 288 and 206 patients, respectively, had tumor assessment data using the mRECIST criteria. Table 2 summarizes the tumor responses of the 494 patients. Although the proportions of patients with CR did not largely differ, those of both ORR and DCR were significantly higher in the longer- than in the shorter-treatment groups (ORR, 47.6% vs. 28.2%; DCR, 92.4% vs. 60.2%; p<0.001 for both). Moreover, the proportion of patients with progressive disease (PD) was significantly lower in the longer-treatment group (7.6% vs. 39.8%; p<0.001).

**Adverse drug reactions.** Table 3 summarizes the ADRs of interest for each group. Overall, the seven ADRs mentioned below (any grade) were more commonly reported in the longer- than in the shorter-treatment groups (p<0.05 for all); >10% higher proportions were found for proteinuria (23.6% vs. 13.1%), HFSR (23.0% vs. 10.0%), hypothyroidism (22.2% vs. 9.4%), and diarrhea (19.6% vs. 8.3%). However, the proportions of patients with any ADRs (any grade) were similar between the two groups (86.9% vs. 82.9%). The occurrence of any ADRs of grade ≥3 was significantly higher in the shorter-treatment group compared to the longer-treatment group (46.4% vs. 38.6%; p = 0.039). The occurrence of the seven ADRs of grade ≥3 was generally similar between the two groups; however, the proportions of fatigue (4.8% vs. 2.8%) and HFSR (3.7% vs. 1.7%) were slightly higher in the longer- than in the shorter-treatment group.

## Baseline patient characteristics of patients with or without dose modification in the longer-treatment group

The patient characteristics characterizing a longer treatment included a younger age, a better ECOG PS, or a lower Child-Pugh score, also lower tumor burden (lower AFP and less high grade VI) (Table 1). More favorable tumor response in patients with longer treatment may have been partly attributed to the high dose of lenvatinib. Therefore, we further analyzed 352 patients in the longer-treatment group to explore the impact of dose modification (RDI <100% due to dose reduction or interruption) on treatment outcomes.

Of 352 patients, 63 (17.9%) patients continued lenvatinib treatment without dose modification during the initial 177 days of treatment (i.e., 177-day RDI ≥100%), whereas 289 (82.1%) patients underwent dose modification during this period (i.e., 177-day RDI <100%) (Fig 1). The distribution of clinical characteristics such as Child-Pugh score and BCLC stage did not significantly differ between the two groups; however, differences were observed in the distribution of body weight and ECOG PS (p<0.05 for both) (Table 4). Approximately half of the patients without dose modification weighed 50 to <60 kg (50.8%), whereas 60 to <70 kg was the most common category (37.4%) in patients with dose modification. The group without

**Table 1.  Baseline patient characteristics according to treatment duration (n = 703).**

| Variables, n (%) | Longer-treatment (≥177 days) (n = 352) | Shorter-treatment (<177 days) (n = 351) | p-value[a] |
|---|---|---|---|
| Sex | | | 0.288 |
| Men | 288 (81.8) | 276 (78.6) | |
| Women | 64 (18.2) | 75 (21.4) | |
| Age, years | | | 0.005* |
| <65 | 67 (19.0) | 50 (14.2) | |
| 65–74 | 156 (44.3) | 127 (36.2) | |
| 75–84 | 111 (31.5) | 144 (41.0) | |
| ≥85 | 18 (5.1) | 30 (8.5) | |
| Body weight, kg | | | 0.105 |
| <50 | 46 (13.1) | 69 (19.7) | |
| ≥50, <60 | 104 (29.5) | 104 (29.6) | |
| ≥60, <70 | 119 (33.8) | 106 (30.2) | |
| ≥70 | 83 (23.6) | 72 (20.5) | |
| Body mass index, kg/m$^2$ | | | 0.322 |
| <18.5 | 22 (6.3) | 35 (10.0) | |
| ≥18.5, <22.0 | 97 (27.6) | 99 (28.2) | |
| ≥22.0, <25.0 | 131 (37.2) | 122 (34.8) | |
| ≥25.0 | 101 (28.7) | 95 (27.1) | |
| Missing | 1 (0.3) | – | |
| ECOG PS | | | <0.001* |
| 0 | 295 (83.8) | 238 (67.8) | |
| 1 | 50 (14.2) | 98 (27.9) | |
| ≥2 | 5 (1.4) | 11 (3.1) | |
| Missing | 2 (0.6) | 4 (1.1) | |
| BCLC stage | | | 0.130 |
| 0 | 1 (0.3) | 1 (0.3) | |
| A | 34 (9.7) | 23 (6.6) | |
| B | 157 (44.6) | 134 (38.2) | |
| C | 152 (43.2) | 180 (51.3) | |
| D | 4 (1.1) | 7 (2.0) | |
| Missing | 4 (1.1) | 6 (1.7) | |
| Portal vein invasion[b] | | | 0.005* |
| Vp0 | 279 (79.3) | 241 (68.7) | |
| Vp1 or 2 | 31 (8.8) | 52 (14.8) | |
| Vp3 or 4 | 34 (9.7) | 48 (13.7) | |
| Unknown | 8 (2.3) | 10 (2.8) | |
| Child-Pugh score | | | <0.001* |
| 5 | 232 (65.9) | 161 (45.9) | |
| 6 | 103 (29.3) | 128 (36.5) | |
| 7 | 8 (2.3) | 41 (11.7) | |
| 8 | 4 (1.1) | 16 (4.6) | |
| 9 | 2 (0.6) | 2 (0.6) | |
| ≥10 | 0 (0.0) | 2 (0.6) | |
| Missing | 3 (0.9) | 1 (0.3) | |
| mALBI grade | | | <0.001* |
| Grade 1 | 135 (38.4) | 81 (23.1) | |

(*Continued*)

**Table 1.** (Continued)

| Variables, n (%) | Longer-treatment (≥177 days) | Shorter-treatment (<177 days) | p-value[a] |
|---|---|---|---|
| | (n = 352) | (n = 351) | |
| Grade 2a | 113 (32.1) | 86 (24.5) | |
| Grade 2b | 100 (28.4) | 158 (45.0) | |
| Grade 3 | 2 (0.6) | 16 (4.6) | |
| Missing | 2 (0.6) | 10 (2.8) | |
| Etiology of HCC[c] | | | –[d] |
| Hepatitis B virus | 81 (23.0) | 47 (13.4) | |
| Hepatitis C virus | 127 (36.1) | 151 (43.0) | |
| Alcohol | 89 (25.3) | 78 (22.2) | |
| NAFLD/NASH | 43 (12.2) | 45 (12.8) | |
| Others | 5 (1.4) | 7 (2.0) | |
| Unknown | 22 (6.3) | 37 (10.5) | |
| History of TKI treatments | | | 0.043* |
| No | 276 (78.4) | 296 (84.3) | |
| Yes | 76 (21.6) | 55 (15.7) | |
| History of TACE | | | 0.132 |
| No | 104 (29.5) | 86 (24.5) | |
| Yes | 248 (70.5) | 265 (75.5) | |
| AFP level, ng/mL | | | <0.001* |
| <200 | 240 (68.2) | 192 (54.7) | |
| ≥200 | 100 (28.4) | 140 (39.9) | |
| Missing | 12 (3.4) | 19 (5.4) | |

[a]Chi-square test

[b]Vp0 = no portal vein invasion, Vp1 = invasion to more distal to the 2nd portal branch; Vp2 = invasion to the 2nd portal branch; Vp3 = invasion to the 1st portal branch; Vp4 = invasion to the main portal vein or contra-lateral portal vein branch.

[c]Patients may be included in multiple categories.

[d]Comparison was not performed because of patients overlapped between categories.

AFP, alpha-fetoprotein; BCLC, Barcelona Clinic Liver Cancer; ECOG PS, Eastern Cooperative Oncology Group performance status; HCC, hepatocellular carcinoma; mALBI grade, modified albumin-bilirubin grade; NAFLD/NASH, non-alcoholic fatty liver disease/non-alcoholic steatohepatitis; TACE, trans-arterial chemoembolization; TKI, tyrosine kinase inhibitor.

dose modifications comprised more patients with a Child-Pugh score of 5 (81.0% vs. 62.6%) and mALBI grade 1 (50.8% vs. 35.6%) than the dose modification group. However, the differences between the two groups were not statistically significant.

## Effectiveness and safety in patients with or without dose modification in the longer-treatment group

In this study, the analysis of tumor assessment using mRECIST criteria was limited due to incomplete data. Specifically, only 49 of 63 patients without dose modification and 239 of 289 patients with dose modification during 177 days were included in the analysis (Fig 1). The tumor response results are summarized in Table 5. Although the difference was not statistically significant, the ORR tended to be higher in patients who did not undergo dose modification than those who underwent dose modification during the 177 days (59.2% vs. 45.2%; p = 0.084). On the other hand, the DCR was similarly high in the respective groups (98.0% vs. 91.2%; p = 0.141).

**Table 2. Tumor response assessed using the mRECIST criteria according to treatment duration (n = 494).**

| Response rate, n (%) | Longer-treatment (≥177 days) | Shorter-treatment (<177 days) | p-value[a] |
|---|---|---|---|
| | (n = 288) | (n = 206) | |
| ORR[a] | 137 (47.6) | 58 (28.2) | <0.001 |
| DCR[b] | 266 (92.4) | 124 (60.2) | <0.001 |
| Best overall response | | | |
| CR | 26 (9.0) | 10 (4.9) | 0.082 |
| PR | 111 (38.5) | 48 (23.3) | <0.001 |
| SD | 129 (44.8) | 66 (32.0) | 0.005 |
| PD | 22 (7.6) | 82 (39.8) | <0.001 |

[a]Fisher's exact test

[b]ORR = proportion of patients with CR and PR

[c]DCR = proportion of patients with CR, PR, and SD

CR, complete response; DCR, disease control rate; mRECIST, modified Response Evaluation Criteria in Solid Tumors; ORR, objective response rate; PD, progressive disease; PR, partial response; SD, stable disease.

Table 6 shows the occurrence of ADRs of interest in all 352 patients in the longer-treatment group by dose modification during 177 days.

## Discussion

Systemic therapy including lenvatinib treatment should ideally be continued for a prolonged period. Using the data from 703 patients from a previous prospective observational study involving Lenvatinib treatment for uHCC in Japan, we examined the characteristics of patients for whom long-term lenvatinib treatment may be considered.

In the present analysis, patients with longer treatment durations tended to have better conditions (e.g., better ECOG PS, lower Child-Pugh score and better mALBi grade) or tumor status (e.g., no portal vein invasion and lower AFP level). These results were consistent with those of previous reports. The absence of portal invasion indicates a less advanced HCCs, which may have been associated with less disease progression and treatment discontinuation, as shown in

**Table 3. The occurrence of ADRs of interest according to treatment duration (n = 703).**

| ADR, n (%) | Longer-treatment (≥177 days) (n = 352) | | Shorter-treatment (<177 days) (n = 351) | |
|---|---|---|---|---|
| | Any grade | Grade ≥3 | Any grade | Grade ≥3 |
| Any ADR[a] | 306 (86.9) | 136 (38.6) | 291 (82.9) | 163 (46.4) |
| Appetite loss | 97 (27.6) | 7 (2.0) | 71 (20.2) | 18 (5.1) |
| Fatigue | 88 (25.0) | 17 (4.8) | 65 (18.5) | 10 (2.8) |
| Hypertension | 88 (25.0) | 25 (7.1) | 62 (17.7) | 20 (5.7) |
| Proteinuria | 83 (23.6) | 24 (6.8) | 46 (13.1) | 23 (6.6) |
| HFSR | 81 (23.0) | 13 (3.7) | 35 (10.0) | 6 (1.7) |
| Hypothyroidism | 78 (22.2) | 3 (0.9) | 33 (9.4) | 4 (1.1) |
| Diarrhea | 69 (19.6) | 8 (2.3) | 29 (8.3) | 11 (3.1) |

[a]Includes any ADR, including but not limited to the listed seven diseases/conditions.

ADR, adverse drug reaction; HFSR, hand-foot skin reaction.

**Table 4. Baseline characteristics of patients with longer treatment by with or without dose modification during the initial 177 days of treatment (n = 352).**

| Variables, n (%) | Patients with longer treatment (≥177 days) | | p-value[a] |
|---|---|---|---|
| | No dose modification (177-day RDI ≥100%) | Dose modification (177-day RDI <100%) | |
| | (n = 63) | (n = 289) | |
| Sex | | | 0.212 |
| Men | 55 (87.3) | 233 (80.6) | |
| Women | 8 (12.7) | 56 (19.4) | |
| Age, years | | | 0.133 |
| <65 | 16 (25.4) | 51 (17.6) | |
| 65–74 | 27 (42.9) | 129 (44.6) | |
| 75–84 | 20 (31.7) | 91 (31.5) | |
| ≥85 | 0 (0.0) | 18 (6.2) | |
| Body weight, kg | | | <0.001* |
| <50 | 6 (9.5) | 40 (13.8) | |
| ≥50, <60 | 32 (50.8) | 72 (24.9) | |
| ≥60, <70 | 11 (17.5) | 108 (37.4) | |
| ≥70 | 14 (22.2) | 69 (23.9) | |
| Body mass index, kg/m$^2$ | | | 0.075 |
| <18.5 | 5 (7.9) | 17 (5.9) | |
| ≥18.5, <22.0 | 25 (39.7) | 72 (24.9) | |
| ≥22.0, <25.0 | 17 (27.0) | 114 (39.4) | |
| ≥25.0 | 16 (25.4) | 85 (29.4) | |
| Missing | – | 1 (0.3) | |
| ECOG PS | | | 0.032* |
| 0 | 59 (93.7) | 236 (81.7) | |
| 1 | 3 (4.8) | 47 (16.3) | |
| ≥2 | 0 (0.0) | 5 (1.7) | |
| Missing | 1 (1.6) | 1 (0.3) | |
| BCLC stage | | | 0.053 |
| 0 | 1 (1.6) | 0 (0.0) | |
| A | 3 (4.8) | 31 (10.7) | |
| B | 24 (38.1) | 133 (46.0) | |
| C | 34 (54.0) | 118 (40.8) | |
| D | 1 (1.6) | 3 (1.0) | |
| Missing | – | 4 (1.4) | |
| Portal vein invasion[a] | | | 0.535 |
| Vp0 | 50 (79.4) | 229 (79.2) | |
| Vp1 or 2 | 4 (6.3) | 27 (9.3) | |
| Vp3 or 4 | 8 (12.7) | 26 (9.0) | |
| Unknown | 1 (1.6) | 7 (2.4) | |
| Child-Pugh score | | | 0.103 |
| 5 | 51 (81.0) | 181 (62.6) | |
| 6 | 11 (17.5) | 92 (31.8) | |
| 7 | 1 (1.6) | 7 (2.4) | |
| 8 | 0 (0.0) | 4 (1.4) | |
| 9 | 0 (0.0) | 2 (0.7) | |
| ≥10 | 0 (0.0) | 0 (0.0) | |
| Missing | – | 3 (1.0) | |

*(Continued)*

**Table 4.** (Continued)

| Variables, n (%) | Patients with longer treatment (≥177 days) | | p-value[a] |
|---|---|---|---|
| | No dose modification (177-day RDI ≥100%) | Dose modification (177-day RDI <100%) | |
| | (n = 63) | (n = 289) | |
| mALBI grade | | | 0.113 |
| Grade 1 | 32 (50.8) | 103 (35.6) | |
| Grade 2a | 19 (30.2) | 94 (32.5) | |
| Grade 2b | 12 (19.0) | 88 (30.4) | |
| Grade 3 | 0 (0.0) | 2 (0.7) | |
| Missing | – | 2 (0.7) | |
| Etiology of HCC[b] | | | –[d] |
| Hepatitis B virus | 22 (34.9) | 59 (20.4) | |
| Hepatitis C virus | 18 (28.6) | 109 (37.7) | |
| Alcohol | 19 (30.2) | 70 (24.2) | |
| NAFLD/NASH | 6 (9.5) | 37 (12.8) | |
| Others | 0 (0.0) | 5 (1.7) | |
| Unknown | 3 (4.8) | 19 (6.6) | |
| History of TKI treatments | | | 0.250 |
| No | 46 (73.0) | 230 (79.6) | |
| Yes | 17 (27.0) | 59 (20.4) | |
| History of TACE | | | 0.100 |
| No | 24 (38.1) | 80 (27.7) | |
| Yes | 39 (61.9) | 209 (72.3) | |
| AFP level, ng/mL | | | 0.839 |
| <200 | 43 (68.3) | 197 (68.2) | |
| ≥200 | 17 (27.0) | 83 (28.7) | |
| Missing | 3 (4.8) | 9 (3.1) | |

[a]Chi-square test.

[b]Vp0 = no portal vein invasion, Vp1 = invasion to more distal to the 2nd portal branch; Vp2 = invasion to the 2nd portal branch; Vp3 = invasion to the 1st portal branch; Vp4 = invasion to the main portal vein or contra-lateral portal vein branch.

[c]Patients may be included in multiple categories.

[d]Comparison was not performed because of patients overlapped between categories.

AFP, alpha-fetoprotein; BCLC, Barcelona Clinic Liver Cancer; ECOG PS, Eastern Cooperative Oncology Group performance status; HCC, hepatocellular carcinoma; mALBI grade, modified albumin-bilirubin grade; NAFLD/NASH, non-alcoholic fatty liver disease/non-alcoholic steatohepatitis; RDI, relative dose intensity; TACE, trans-arterial chemoembolization; TKI, tyrosine kinase inhibitor.

[18]. Regarding TKI history, the patients with this treatment history received lenvatinib treatment for an extended period. Two cases were considered for these patients. TKI-experienced patients might have been used to AEs commonly experienced in TKI treatments, and the previous experience with AEs may have avoided early discontinuation due to AEs. On the other hand, TKI-tolerable patients might have been involved in this study. Patients with better liver functional reserve [6, 19], younger age [19], or better ECOG PS [6], are less likely to discontinue treatments due to serious AEs, and these characteristics may allow for longer treatment.

In the previous studies, the association between baseline liver functional reserve and treatment duration was also reported [7, 8, 20]. In other words, these reports imply that careful management is required for patients with a lower functional reserve or poor overall status at baseline. Although possible confounding factors exist and these variables do not necessarily

**Table 5. Tumor response by with or without dose modification during the initial 177 days of treatment in patients with longer treatment (n = 288[a]).**

| Response rate, n (%) | No dose modification (177-day RDI ≥100%) | Dose modification (177-day RDI <100%) | p-value[b] |
|---|---|---|---|
| | (n = 49) | (n = 239) | |
| ORR[b] | 29 (59.2) | 108 (45.2) | 0.084 |
| DCR[c] | 48 (98.0) | 218 (91.2) | 0.141 |
| Best overall response | | | |
| CR | 4 (8.2) | 22 (9.2) | 1.000 |
| PR | 25 (51.0) | 86 (36.0) | 0.054 |
| SD | 19 (38.8) | 110 (46.0) | 0.430 |
| PD | 1 (2.0) | 21 (8.8) | 0.141 |

[a]Of 352 patients with ≥177 days of treatment, 288 patients with tumor assessment data using the mRECIST criteria were analyzed.

[b]Fisher's exact test

[c]ORR = proportion of patients with CR and PR

[d]DCR = proportion of patients with CR, PR, and SD

CR, complete response; DCR, disease control rate; mRECIST, modified Response Evaluation Criteria in Solid Tumors; ORR, objective response rate; PD, progressive disease; PR, partial response; RDI, relative dose intensity; SD, stable disease.

determine the treatment duration, the observed patient profiles would help physicians plan a treatment strategy for candidates of lenvatinib treatment. As expected, patients who received a longer treatment had a significantly higher ORR than those who received a shorter treatment, which probably reflected fewer initial treatment failures in these patients. Given that the objective response is an independent predictor of OS in patients treated with lenvatinib for uHCC [10], patients who successfully responded to treatment, thus continued treatment for extended period, are expected to have increased survival benefits due to longer tumor control. Indeed, our exploratory analyses showed that patients who achieved an objective response (CR or PR) had a significantly longer treatment duration than those who achieved SD or PD, and these patients also had a significantly longer OS (S1 Fig).

In our analysis of patients who received longer treatment, the ORR was numerically higher, although not statistically significant in patients without dose modification than in those with

**Table 6. The occurrence of ADRs of interest by with or without dose modification during the initial 177 days of treatment in patients with longer treatment (n = 352).**

| ADR, n (%) | No dose modification (177-day RDI ≥100%) | | Dose modification (177-day RDI <100%) | |
|---|---|---|---|---|
| | (n = 63) | | (n = 289) | |
| | Any grade | Grade ≥3 | Any grade | Grade ≥3 |
| Any ADR[a] | 43 (68.3) | 10 (15.9) | 263 (91.0) | 126 (43.6) |
| Appetite loss | 10 (15.9) | 2 (3.2) | 87 (30.1) | 5 (1.7) |
| Fatigue | 12 (19.0) | 1 (1.6) | 76 (26.3) | 16 (5.5) |
| Hypertension | 16 (25.4) | 6 (9.5) | 72 (24.9) | 19 (6.6) |
| Proteinuria | 4 (6.3) | 0 (0.0) | 79 (27.3) | 24 (8.3) |
| HFSR | 11 (17.5) | 0 (0.0) | 70 (24.2) | 13 (4.5) |
| Hypothyroidism | 10 (15.9) | 0 (0.0) | 68 (23.5) | 3 (1.0) |
| Diarrhea | 6 (9.5) | 0 (0.0) | 63 (21.8) | 8 (2.8) |

[a]Includes any ADR, including but not limited to the listed seven diseases/conditions.

ADR, adverse drug reaction; HFSR, hand-foot skin reaction; RDI, relative dose intensity.

dose modification. However, the DCR was equivalently high regardless of dose modification (98.0% vs. 91.2%). The lack of a significant difference between the groups was probably because the comparison was made among patients who received longer treatment. Nevertheless, the result also indicate that treatment should be continued, even if the dose is adjusted to manage AEs [21]. Indeed, in our overall population, the mean RDI gradually declined to approximately 60% in the first few months and was sustained thereafter (S2 Fig). These findings imply that patients benefit from dose modifications, and the adjustments may lead to favorable treatment outcomes. In this study, drug discontinuation in our overall population occurred for the following reasons (which were mutually inclusive): AEs accounted for 43.7% of cases including those resulting in death, ADRs for 39.5% including those resulting in death, and PD for 32.3%. ADRs leading to treatment discontinuations included appetite loss (6.3%), fatigue (4.7%), proteinuria (3.3%) (S1 Table). Given these findings, it is recommended that these ADRs should be managed with caution to prolong treatment.

When we additionally explored the impact of dose modification within 8 weeks of treatment in 423 patients, the ORR was significantly higher in patients who did not undergo dose modification during the initial 8 weeks (52.4% vs. 39.7%, p = 0.018; S2 Table). This result is in accordance with the findings of previous small retrospective studies, suggesting that maintaining a high RDI during the initial treatment phase (e.g., 8 weeks [11, 12] or approximately 30 days [13–15]) is important for achieving a more favorable tumor response. Additionally, these results may also indicate that, especially during the initial treatment phase (e.g. 8 weeks), administering a recommended dose to achieve an objective response, which leads to a prolonged OS [10], is an important treatment strategy.

Finding a balance among patient characteristics including liver functional reserve, treatment effects, and AE management is crucial for effective, long-term treatment with lenvatinib. In our subgroup analysis by dose modification during the initial 177 days in patients with longer treatment, body weight of 50 to <60 kg, an ECOG PS of 0, and a Child-Pugh score of 5 were suggested as patient characteristics that can expect higher tolerability. Better liver functional reserve and overall status may suggest a lower risk of toxicity [22] and AEs. Given that the body weight-based standard dose of lenvatinib uses the cut-off of 60 kg [23], patients weighing just below 60 kg may tolerate a dose higher than their standard dose of 8 mg/day. Recent studies reported that skeletal muscle mass might be a good predictor of tolerability and prognosis in cases of lenvatinib treatment [24, 25]. It may be advisable to tailor an optimal dose, considering the patient's body weight or other factors to maximize treatment effectiveness for patients with borderline body weight.

The present post-hoc analysis had several limitations that were mainly inherent to the original observational study [6]. First, generalizability is limited due to the inclusion of patients treated only in Japan, where treatment patterns may differ from those in other countries. Second, possible confounding factors should be considered when interpreting the results. For example, a better tumor response in patients who did not receive dose modification in the initial eight weeks (additional analysis) might have been due to better liver functional reserve or other clinical factors, which can also affect dose intensity. Thus, the causal relationship between dose intensity and tumor response remains to be confirmed in future studies. Third, tumor response was not evaluated using uniform criteria in routine practice settings, which reduced the number of patients included for the effectiveness evaluation. Also, the evaluation was performed at each institution, instead of in an independent central review, which may have lowered the accuracy of the tumor evaluations. Lastly, as the study was conducted in the early days of lenvatinib approval, the treatment patterns in this analysis probably reflect the conservative use of this new agent, which may differ from those used today.

## Conclusion

In this study, we have clarified the characteristics of patients receiving long-term treamtment of lenvatinib and the dosage in clinical practice in Japan. As previously reported, patients with longer treatment were more likely to have a better ECOG PS and a mALBI grade, and lower Child-Pugh score.

Although systemic lenvatinib treatment should ideally be continued for a prolonged period, patients receiving longer treatment are more likely to experience ADRs due to extended drug exposure. Appropriate dose adjustments at the physician's discretion may contribute to long-term continuation.

## Supporting information

**S1 Table. The occurrence of ADRs[a] that lead to treatment discontinuation (n = 703).** [a]The proportion of ADRs ≥1% are presented in the table. ADR, adverse drug reaction; HFSR, hand-foot skin reaction.
(DOCX)

**S2 Table. Tumor response by with or without dose modification during the initial eight weeks of treatment in patients with ≥8 weeks of treatment (n = 423[a]).** [a]Of 703 patients, 550 patients had ≥8 weeks of treatment. Of these, 170 patients had no dose modification and 380 had dose modification, during the initial eight weeks of treatment. Of these, 126 and 297 patients (423 in total) with tumor assessment data using the mRECIST criteria were analyzed. [b]Fisher's exact test, [c]ORR = proportion of patients with CR and PR, [d]DCR = proportion of patients with CR, PR, and SD, CR, complete response; DCR, disease control rate; mRECIST, modified Response Evaluation Criteria in Solid Tumors; ORR, objective response rate; PD, progressive disease; PR, partial response; RDI, relative dose intensity; SD, stable disease.
(DOCX)

**S1 Fig. Kaplan-Meier curves for (a) treatment duration and (b) overall survival according to treatment response (CR or PR vs. SD or PD).** (a) Treatment duration was significantly longer in patients with objective response (CR or PR) than in those without an objective response (SD or PD). (b) Overall survival was significantly longer in patients with objective response (CR or PR) than in those without an objective response (SD or PD). CR, complete response; PD, progressive disease; PR, partial response; SD, stable disease.
(DOCX)

**S2 Fig. The transition of mean RDIs per two weeks in the overall study population (n = 703).** The RDI was calculated as the ratio of the total dosage delivered to the body weight-based standard dosage (i.e., the treatment duration [day] multiplied by 12 mg or 8 mg, depending on body weight). The mean RDI was around 80% initially, then gradually declined to approximately 60% in a few months, and sustained at the level after that. RDI, relative dose intensity.
(DOCX)

## Acknowledgments

The medical writing support was provided by Clinical Study Support, Inc. (Nagoya, Japan), under contract with Eisai Co., Ltd.

## Author Contributions

**Conceptualization:** Natsumi Suzuki, Katsuaki Motoyoshi, Wanjun Zhu.

**Formal analysis:** Natsumi Suzuki, Wanjun Zhu.

**Supervision:** Katsuaki Motoyoshi.

**Writing – original draft:** Katsuaki Motoyoshi.

**Writing – review & editing:** Tatsuya Yamashita, Natsumi Suzuki, Wanjun Zhu, Junji Furuse.

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
