## [Decision Letter · Decision Letter 0]

25 Oct 2023

PONE-D-23-28810Characteristics of patients with longer treatment period of lenvatinib for unresectable hepatocellular carcinoma: A post-hoc analysis of post-marketing surveillance study in JapanPLOS ONE

Dear Dr. Yamashita,

Thank you for submitting your manuscript to PLOS ONE. After careful consideration, we feel that it has merit but does not fully meet PLOS ONE’s publication criteria as it currently stands. Therefore, we invite you to submit a revised version of the manuscript that addresses the points raised during the review process.

We look forward to receiving your revised manuscript.

Kind regards,

Jin-Yu Sun

Academic Editor

PLOS ONE

 [This study was funded by Eisai Co., Ltd. The sponsor was involved in the study design, data collection and analysis, decision to publish, and preparation of the manuscript.]. 

[I have read the journal's policy and the authors of this manuscript have the following competing interests:

Tatsuya Yamashita has received travel grants and honoraria for speaking or participation at meetings from Chugai Pharma, Eli Lilly Japan, and Eisai.

Natsumi Suzuki, Katsuaki Motoyoshi, and Wanjun Zhu are employees of Eisai Co., Ltd.

Junji Furuse received research grants from Astellas, Astra Zeneca, Incyte Biosciences Japan, Eisai, MSD, Ono Pharmaceutical, Sanofi, J-Pharma, Daiichi Sankyo, Sumitomo Dainippon, Taiho Pharmaceutical, Takeda, Delta-Fly-Pharma, Chugai Pharma. He has received travel grants and honoraria for speaking or participation at meetings from Ono Pharmaceutical, Chugai Pharma, Incyte Biosciences Japan, Eisai, Eli Lilly Japan, Astra Zeneca, Yakult Honsha, Servier Japan, MSD, Novartis Pharma, Takeda, Bayer, Taiho Pharmaceutical, EA Pharma, Teijin pharma, Daiichi Sankyo, Terumo, Fuji film, Astellas, Onco Therapy Science, Delta-Fly-Pharma, Merck Bio, Incyte Biosciences Japan, J-Pharma.].

Additional Editor Comments:

Based on the comments of the two reviewers, a major revision is required before further consideration. Please revise the manuscript carefully.

Reviewers' comments:

Reviewer's Responses to Questions

**Comments to the Author**

1. Is the manuscript technically sound, and do the data support the conclusions?

Reviewer #1: Yes

Reviewer #2: Partly

2. Has the statistical analysis been performed appropriately and rigorously? 

Reviewer #1: Yes

Reviewer #2: N/A

3. Have the authors made all data underlying the findings in their manuscript fully available?

Reviewer #1: Yes

Reviewer #2: Yes

4. Is the manuscript presented in an intelligible fashion and written in standard English?

Reviewer #1: Yes

Reviewer #2: Yes

5. Review Comments to the Author

Reviewer #1: This article by Yamashita et al. analyzes real-world clinical data of lenvatinib treatment in patients with unresectable hepatocellular carcinoma. The authors focused on longer lenvatinib treatment group and compared characteristics with those of shorter treatment. They also analyzed the impact of dose modification on the efficacy and adverse events in longer treatment group. The reviewer believes that this paper provides important real-world data that will benefit clinicians.

Please consider the following point.

1. the authors did not perform statistical analyses of the longer vs. shorter treatment and dose-modified vs. non-modified groups. The reviewer believe that it is a rigorous method not to make statistical comparisons in post-hoc analyses. However, there are several points where statistical comparisons can be made, such as comparing any ADR or radiological response between patients experienced dose modification within 177 days and those did not. Please comment on this point.

Reviewer #2: T. Yamashita et al. have reported about a post-hoc analysis of post-marketing surveillance study in Japan. To establish the position of LEN in the present guidelines, it is also very important to investigate about post-hoc analysis about LEN in Japanese population. However, there are some concerns in this study to be published as an original article.

#1. There is no data provided for the statistical analysis in the document and Tables. It would be helpful if the authors could explain the reason for this omission.

#2. If possible, authors may add the data about the reason why LEN was discontinued.　Is it progressive disease, discontinuation due to ADR, or post-LEN therapy (e.g. TACE) ?

#3. While the median treatment duration is mentioned for the entire cohort, it would be beneficial to provide the minimum and maximum values for treatment duration within both longer and shorter treatment groups, respectively.

#4. It's noted that the Relative Dose Intensity (RDI) cut-off is set at 100%. Have the authors explored other RDI cut-off values?

#5. Is there any available data regarding the post-LEN treatment administered to the patients in the study?

#6. Have the authors investigated the longitudinal changes of hepatic function such as Child-Pugh, ALBI, etc. at different time points (e.g., pre-treatment, 4 weeks, 8 weeks, and 12 weeks)? Which show differences in longer vs. shorter?

I am interested in whether liver function of longer-treatment patients to be better maintained.

6. PLOS authors have the option to publish the peer review history of their article (what does this mean?). If published, this will include your full peer review and any attached files.

Reviewer #1: No

Reviewer #2: No

---

## [Author Response · Author response to Decision Letter 0]

10 Dec 2023

Reviewer 1

This article by Yamashita et al. analyzes real-world clinical data of lenvatinib treatment in patients with unresectable hepatocellular carcinoma. The authors focused on longer lenvatinib treatment group and compared characteristics with those of shorter treatment. They also analyzed the impact of dose modification on the efficacy and adverse events in longer treatment group. The reviewer believes that this paper provides important real-world data that will benefit clinicians.

Please consider the following point.

#1 the authors did not perform statistical analyses of the longer vs. shorter treatment and dose-modified vs. non-modified groups. The reviewer believe that it is a rigorous method not to make statistical comparisons in post-hoc analyses. However, there are several points where statistical comparisons can be made, such as comparing any ADR or radiological response between patients experienced dose modification within 177 days and those did not. Please comment on this point.

[Response]

Your feedback is greatly valued, and we fully understand the importance of including a statistical analysis. In our initial work, we did not carry out a statistical examination due to concerns regarding selection bias among subgroups. However, in the revised version, we have incorporated a thorough statistical analysis into our study. Please kindly examine the Figures and Tables, to gauge the impact of these improvements on our findings. 

Reviewer 2

T. Yamashita et al. have reported about a post-hoc analysis of post-marketing surveillance study in Japan. To establish the position of LEN in the present guidelines, it is also very important to investigate about post-hoc analysis about LEN in Japanese population. However, there are some concerns in this study to be published as an original article.

#1 . There is no data provided for the statistical analysis in the document and Tables. It would be helpful if the authors could explain the reason for this omission.

[Response]

Your feedback is greatly valued, and we fully understand the importance of including a statistical analysis. In our initial work, we did not carry out a statistical examination due to concerns regarding selection bias among subgroups. However, in the revised version, we have incorporated a thorough statistical analysis into our study. Please kindly examine the Figures and Tables, to gauge the impact of these improvements on our findings. 

#2. If possible, authors may add the data about the reason why LEN was discontinued. Is it progressive disease, discontinuation due to ADR, or post-LEN therapy (e.g. TACE) ?

[Response]

We appreciate your comment. We have included the reason for the discontinuation in the manuscript, and we kindly request that you review it for further details (Line 359–362 in the Revised manuscript with track changes). 

#3. While the median treatment duration is mentioned for the entire cohort, it would be beneficial to provide the minimum and maximum values for treatment duration within both longer and shorter treatment groups, respectively.

[Response]

We appreciate your suggestion. We have now incorporated the minimum and maximum values (ranging from 2 days to 482 days) into the text, and those in 2 cohorts (Longer: range 177–482 days, Shorter: range 2–174 days) and we kindly ask you to verify this inclusion (Line 168 and Line 171–172). 

#4. It's noted that the Relative Dose Intensity (RDI) cut-off is set at 100%. Have the 　authors explored other RDI cut-off values?

[Response]

We appreciate your comments, and we've given careful consideration to this matter. With respect to the cutoff value, it is noteworthy that previous reports commonly mention an 8-week RDI. However, there remains uncertainty about which specific time point for RDI is most appropriate or whether RDI is a suitable indicator, given the limited evidence available. In light of this context, our post-hoc analysis examined the results by categorizing them into two groups: "Full dose" and "Dose modification," using the median administration duration as the time point. 

#5. Is there any available data regarding the post-LEN treatment administered to the patients in the study ?

[Response]

Thank you for the confirmation.

Since this PMS was designed to assess the safety and dosing of lenvatinib, we did not collect information on post-treatment.

#6. Have the authors investigated the longitudinal changes of hepatic function such as Child-Pugh, ALBI, etc. at different time points (e.g., pre-treatment, 4 weeks, 8 weeks, and 12 weeks)? Which show differences in longer vs. shorter?

I am interested in whether liver function of longer-treatment patients to be better maintained.

[Response]

Thank you for the confirmation.

Since this PMS was designed to assess the safety and dosing of lenvatinib, we did not collect information on the longitudinal changes of hepatic function.

---

## [Decision Letter · Decision Letter 1]

25 Jan 2024

Characteristics of patients with longer treatment period of lenvatinib for unresectable hepatocellular carcinoma: A post-hoc analysis of post-marketing surveillance study in Japan

PONE-D-23-28810R1

Dear Dr. Yamashita,

We’re pleased to inform you that your manuscript has been judged scientifically suitable for publication and will be formally accepted for publication once it meets all outstanding technical requirements.

Kind regards,

Jin-Yu Sun

Academic Editor

PLOS ONE

Additional Editor Comments (optional):

Reviewers' comments:

Reviewer's Responses to Questions

**Comments to the Author**

1. If the authors have adequately addressed your comments raised in a previous round of review and you feel that this manuscript is now acceptable for publication, you may indicate that here to bypass the “Comments to the Author” section, enter your conflict of interest statement in the “Confidential to Editor” section, and submit your "Accept" recommendation.

Reviewer #2: All comments have been addressed

2. Is the manuscript technically sound, and do the data support the conclusions?

Reviewer #2: Yes

3. Has the statistical analysis been performed appropriately and rigorously? 

Reviewer #2: Yes

4. Have the authors made all data underlying the findings in their manuscript fully available?

Reviewer #2: Yes

5. Is the manuscript presented in an intelligible fashion and written in standard English?

Reviewer #2: Yes

6. Review Comments to the Author

Reviewer #2: The author responded to the reviewers' comments appropriately.

Now this manuscript seems to be acceptable.

7. PLOS authors have the option to publish the peer review history of their article (what does this mean?). If published, this will include your full peer review and any attached files.

Reviewer #2: No

---

## [Editor Report · Acceptance letter]

28 Feb 2024

PONE-D-23-28810R1 

PLOS ONE

Dear Dr. Yamashita, 

I'm pleased to inform you that your manuscript has been deemed suitable for publication in PLOS ONE. Congratulations! Your manuscript is now being handed over to our production team.

Kind regards, 

on behalf of

Dr. Jin-Yu Sun 

Academic Editor

PLOS ONE